# Qualitative and Semiquantitative Parameters of ^18^F-FDG-PET/CT as Predictors of Malignancy in Patients with Solitary Pulmonary Nodule

**DOI:** 10.3390/cancers15041000

**Published:** 2023-02-04

**Authors:** Ferdinando Corica, Maria Silvia De Feo, Maria Lina Stazza, Maria Rondini, Andrea Marongiu, Viviana Frantellizzi, Susanna Nuvoli, Alessio Farcomeni, Giuseppe De Vincentis, Angela Spanu

**Affiliations:** 1Department of Radiological Sciences, Oncology and Anatomo-Pathology, Sapienza University of Rome, 00185 Rome, Italy; 2Unit of Nuclear Medicine, Department of Medicine, Surgery and Pharmacy, University of Sassari, 07100 Sassari, Italy; 3Department of Economics & Finance, University of Rome “Tor Vergata”, 00133 Rome, Italy

**Keywords:** ^18^F-FDG, solitary pulmonary nodule, PET, SUVmax, SUVmean, TLG, MTV

## Abstract

**Simple Summary:**

This study aims to evaluate the reliability of qualitative and semiquantitative parameters of ^18^F-FDG PET-CT, and eventually a correlation between them, in predicting the risk of malignancy in patients with solitary pulmonary nodule (SPN) before the diagnosis of lung cancer. Qualitative and semiquantitative parameters can be considered reliable tools in patients with SPN, since cut-offs for SUVmax, SUVmean, TLG and MTV showed good sensitivity and specificity in predicting malignancy.

**Abstract:**

This study aims to evaluate the reliability of qualitative and semiquantitative parameters of ^18^F-FDG PET-CT, and eventually a correlation between them, in predicting the risk of malignancy in patients with solitary pulmonary nodules (SPNs) before the diagnosis of lung cancer. A total of 146 patients were retrospectively studied according to their pre-test probability of malignancy (all patients were intermediate risk), based on radiological features and risk factors, and qualitative and semiquantitative parameters, such as SUVmax, SUVmean, TLG, and MTV, which were obtained from the FDG PET-CT scan of such patients before diagnosis. It has been observed that visual analysis correlates well with the risk of malignancy in patients with SPN; indeed, only 20% of SPNs in which FDG uptake was low or absent were found to be malignant at the cytopathological examination, while 45.45% of SPNs in which FDG uptake was moderate and 90.24% in which FDG uptake was intense were found to be malignant. The same trend was observed evaluating semiquantitative parameters, since increasing values of SUVmax, SUVmean, TLG, and MTV were observed in patients whose cytopathological examination of SPN showed the presence of lung cancer. In particular, in patients whose SPN was neoplastic, we observed a median (MAD) SUVmax of 7.89 (±2.24), median (MAD) SUVmean of 3.76 (±2.59), median (MAD) TLG of 16.36 (±15.87), and a median (MAD) MTV of 3.39 (±2.86). In contrast, in patients whose SPN was non-neoplastic, the SUVmax was 2.24 (±1.73), SUVmean 1.67 (±1.15), TLG 1.63 (±2.33), and MTV 1.20 (±1.20). Optimal cut-offs were drawn for semiquantitative parameters considered predictors of malignancy. Nodule size correlated significantly with FDG uptake intensity and with SUVmax. Finally, age and nodule size proved significant predictors of malignancy. In conclusion, considering the pre-test probability of malignancy, qualitative and semiquantitative parameters can be considered reliable tools in patients with SPN, since cut-offs for SUVmax, SUVmean, TLG, and MTV showed good sensitivity and specificity in predicting malignancy.

## 1. Introduction

A solitary pulmonary nodule (SPN) is defined as a well-circumscribed round lesion measuring up to 3 cm in diameter and surrounded by an aerated lung. It is a common radiologic finding that is often discovered incidentally and may require significant workup to establish a definitive diagnosis [1]. SPNs are documented in 8 to 51% of all chest computed tomographic (CT) scans and are considered separately from pulmonary lesions accompanied by additional radiographic abnormalities (e.g., atelectasis, pleural effusion, or enlarged intrathoracic lymph nodes). The reported incidence of lung cancer in patients with SPN varies widely, from 2–13% in screening studies to 46–82% in positron-emission tomography (PET) studies [2,3]. In a recent meta-analysis, ^18^F-FDG-PET/CT was shown to have good diagnostic performance in SPNs’ differentiation, yielding a sensitivity of 82% and a specificity of 81% [4].

The characterization of SPNs in FDG-PET is based on qualitative and semiquantitative parameters [5]. Qualitative assessment of FDG uptake is based on the visual evaluation of the lesion compared with normal FDG uptake in the surrounding background or with the reference FDG uptake in the mediastinal blood pool or the liver. Semiquantitative parameters include the standardized uptake value (SUV), which is commonly reported either as the maximum (SUVmax) or as the mean (SUVmean) value of all voxels within an ROI [6], and volumetric parameters such as metabolic tumor volume (MTV), defined as the total number of voxels within a volume of interest that have uptake above a predetermined SUV threshold, and total lesion glycolysis (TLG), calculated as MTV × SUVmean [7].

This retrospective study aims to evaluate and correlate quantitative and semiquantitative data of ^18^F-FDG-PET/CT in patients with SPN in order to predict the risk of malignancy of such nodules. Indeed, while most of the diagnostic accuracy studies are limited by the use of a single threshold (e.g., standardized uptake value [SUV] = 2.5) for distinguishing malignant from benign nodules [8]), in this study, each parameter, either qualitative or semiquantitative, is evaluated individually and in correlation with the others in order to identify and propose optimal cut-offs to determine the risk of malignancy for semiquantitative data, such as SUVmax, SUVmean, TLG, MTV, and to correlate the pre-test probability of malignancy with qualitative and semiquantitative data at ^18^F-FDG-PET/CT scan. The study was carried out in a selected series of patients with a pre-test intermediate likelihood of malignancy.

## 2. Materials and Methods

### 2.1. Patient Population

The present study was a retrospective evaluation of 146 consecutive patients (49 females, 97 males, with a mean age of 68.23 ± 8.32 years) with SPN who underwent ^18^F-FDG-PET/CT during November 2014 and February 2021.

All patients had already undergone computed tomography (CT) scans of the chest and were referred to ^18^F-FDG-PET/CT for lesion metabolic activity evaluation. All patients had a pre-test intermediate likelihood of malignancy. The pre-test probability of malignancy was calculated according to Brock model, taking into account both clinical and radiological findings [9]. Patients with a low or high pre-test likelihood of malignancy were excluded from the study.

The main demographic data and known cancer risk factors in the patient series, such as smoking history, occupational exposure, familiarity with lung cancer, previous cancer, TBC during childhood, and pulmonary disease, as well as the dimension of the SPNs at CT scan and SPNs’ lung localization (superior, middle, or inferior lobe) are illustrated in Table 1. The malignant or benign nature of any SPN was assessed by the gold-standard cytopathological evaluation of obtained specimens.

### 2.2. ^18^F-FDG PET/CT Acquisition and Analysis

All patients underwent an ^18^F-FDG-PET/CT examination at the Unit of Nuclear Medicine of the University of Sassari using a Discovery 710 PET/CT scanner (GE Healthcare). Before the scan, all patients fasted for at least 6 h and were checked for blood glucose levels < 150 mg/dL.

A whole-body PET/CT scan was performed in 3D mode, according to routine clinical protocol, starting approximately 60 min after the intravenous injection of 3.75 MBq/Kg of the radiotracer. PET images were acquired in helicoidal mode and reconstructed with an iterative reconstruction algorithm (GE-VPFXS). CT images were also acquired in helicoidal mode, with 120 kVp tube voltage and automatically adjusted tube current, and then utilized for attenuation correction and anatomic localization. Images were read by two well-trained local nuclear medicine physicians with more than 15 years of experience in imaging interpretation. Disagreement was resolved by consensus.

PET images were analyzed both qualitatively and semiquantitatively. Qualitative analysis of FDG uptake in SPNs was carried out according to the visual Herder 4-point scoring system [10], as 1 = intense (uptake markedly higher than mediastinal blood pool), 2 = moderate (uptake higher than mediastinal blood pool), 3 = faint (uptake ≤ mediastinal blood pool), and 4 = absent. Furthermore, semiquantitative analysis was performed in a dedicated console (Advantage Workstation Volume Share 7; General Electric) drawing an isocontour volume of interest (VOI) in the SPN and extrapolating SUVmax, SUVmean, TLG, and MTV.

### 2.3. Statistical Analysis

The target was diagnosed with a cytopathological examination. Data are expressed as mean and standard deviation or median and MAD (median absolute deviation), as appropriate. Receiver Operating Characteristic (ROC) curves and the related Area Under the Curve (AUC) were computed for each of several possible quantitative variables. Optimal cut-offs were set as the ones maximizing the sum of sensitivity and specificity. Spearman correlations were computed to assess the association among variables. A logistic regression analysis was performed to test the association between the variables and the risk of malignity.

## 3. Results

Among the 146 patients who underwent ^18^F-FDG-PET/CT, 91/146 (62.33%) had a primary lung cancer and the remaining 55/146 (37.67%) had benign lesions. The definitive diagnosis of SPNs is reported in Table 2.

Among the variables considered (age, smoking history, occupational exposure, familiarity with lung cancer, previous cancer, TBC during childhood COPD, pulmonary fibrosis, emphysema, nodule lung localization, and nodule size), only age (OR = 1.05, (95% CI: 1.01–1.11, *p* = 0.00875) and nodule size (OR = 1.1; 95% CI: 1.05–1.18, *p* = 0.00062) have proven significant predictors for malignancy at logistic regression analysis.

In addition, as regards the size of the nodules, this was 19.77 ± 6.17 in malignant nodules and 15.81 ± 6.41 in benign ones, and 22.75 mm proved to be the best cut-off in terms of predictive value with a sensitivity of 37.4% and a specificity of 87%. Moreover, nodule size correlated significantly with the intensity of FDG uptake at visual analysis (r = 0.338, *p* < 0.001) and with SUVmax (r = 0.419, *p* < 0.001). ROC analysis with AUC of nodule size is reported in Figure 1.

Visual analysis revealed that FDG uptake was low or absent in 20 of the overall patients (13.69%), moderate in 44 (30.13%) patients, and intense in 82 (56.16%) patients. Among the patients with SPN in which FDG uptake was low or absent, histological examination was positive for malignancy in 4 cases (20%); SPNs with moderate FDG uptake were positive in 20 cases (45.45%) while SPNs with intense uptake were positive for malignancy in 74 cases (90.24%) The visual analysis findings observed in the overall population and in malignant and non malignant SPNs are reported in Table 3.

Regarding to the results of the FDG-PET semiquantitative analysis, all parameters showed higher median (MAD) values in malignant SPNs than in benign ones. Median (MAD) values of SUVmax, SUVmin, TLG, and MTV, together with the optimal cut-offs in terms of the predictive values of malignancy, obtained in the overall population and in malignant and non malignant SPNs are illustrated in Table 4.

ROC analysis with AUC of SUVmax, SUVmean, MTV, and TLG is reported in Figure 2, while the sensitivity and specificity values of these semiquantitative parameters derived by using the optimal cut-offs are reported in Table 5.

## 4. Discussion

The current study, carried out in a selected series of patients with a pre-test intermediate likelihood of malignancy, shows how FDG-PET semiquantitative parameters can play an important role in predicting the malignancy of SPNs.

Among semiquantitative parameters, SUVmax is the most commonly used in clinical practice and the threshold value of 2.5 continues to be the most employed reference standard in differentiating malignant from benign SPNs. Additionally, recently, Erdogdu et. al., in a series of 223 patients with SPNs undergoing pulmonary resection, demonstrated that among several clinic and radiological features, such as age, smoking status, and pack years of smoking, SUVmax, solid component of nodules, spiculation, pleural tag, lobulation, calcification, and higher density, only SUVmax greater than 2.5, spiculation, and age older than 61 years are significant predictors for malignancy in patients with SPNs [11].

In an equally recent multicenter study that involved 355 patients, Weir-McCall et al. stated that SUVmax is the most accurate technique for the diagnosis of solitary pulmonary nodules; however, they have also demonstrated that the diagnostic thresholds should be altered according to nodule size, setting optimal cut-offs for SUVmax according to the lesion size: 1.75 for lesions <12 mm; 2.55 for lesions ranging between 12 and 16 mm; and 3.6 for lesions > 16 mm [12]. In a previous monocentric study, carried out in a series of 88 patients, ROC curve shows SUVmax >3.635 as the best threshold with a sensitivity of 83.3% and a specificity 62.5% in differentiating malignant from benign pulmonary nodules [13]. The above results are consistent with our data that demonstrated a significant correlation between nodule size and SUVmax and a value of 3.625 as the optimal cut-off, with 86.6% sensitivity and 69.1% specificity. Moreover, in our series, nodule size of 22.75 mm proved to be the best cut-off in terms of predictive value. This value was associated with a high specificity (87%); however, sensitivity was low (37.4%), thus suggesting a limited diagnostic effectiveness of the above cut-off in differentiating malignant from benign SPNs when considered alone.

The other metabolic parameters taken into account in the current study are TLG and MTV. In general, since SUVmax is considered the most accurate semiquantitative parameter in the diagnosis of SPN, there is a lack of studies including the reliability of MTV and TLG in ^18^F-FDG PET/CT [14,15]. In a retrospective study, Erdogan et al. found cut-off values for TLG (31.88 g; sensitivity 76.6%, specificity 89.5%, accuracy 78.8%) and MTV (20 mL; sensitivity 79.8%, specificity 68.4%, accuracy 77.9%) in solid nodules, emphasizing the correlation between lesion diameter and semiquantitative data, and stating that MTV could be a more reliable parameter than SUVmax in smaller nodules [16]. This study aims to investigate the probability of malignancy in patients with SPN based on the morphological characteristics of the nodules and the qualitative and semiquantitative data of such nodules at ^18^F-FDG-PET/CT. As stated earlier, patient stratification in this sample was based on the pre-test risk of malignancy including only patients with an intermediate pre-test risk of malignancy, differently from most multicenter studies available in the literature which consider more heterogeneous population samples without pre-test likelihoods of malignancy selection. This difference makes our study more selective and homogeneous, with results with less bias. Evangelista et al. carried out an Italian retrospective multicenter trial involving 502 patients with SPNs stratified according to the likelihood of malignancy [8]. However, their results are not comparable with those of the current study since different semiquantitative FDG-PET/TC parameters were used.

Qualitative analysis based on FDG uptake intensity is reliable in evaluating the malignancy of SPN, since we observed that in our group of patients a high FDG uptake correlates with malignancy in 90.45% of cases. Moreover, we have demonstrated that nodule size correlate significantly with the intensity of FDG uptake at visual analysis. Furthermore, it has been observed how increasing values of SUVmax, TLG, and MTV strongly correlate with a high probability of malignancy in SPNs. These data are supported by the observation that TLG increases linearly with respect to SUVmax and MTV in neoplastic nodules, suggesting how TLG can be considered a reliable parameter, when used in combination with SUVmax, in assessing the probability of malignancy before the cytopathological examination [17]. This also reflects how the use of quantitative methods in both CT and PET/CT images of SPNs can be crucial in the discrimination between benign and malignant lung nodules compared with qualitative examination alone [18,19]. Indeed, it has been already observed how benign nodules on CT scans tend to be flatter than the malignant ones, while the malignant ones have a more isotropic morphology [20]. Confirming our observations, it was also stated that the radiomics features from PET showed that radiotracer uptake (SUVmin, SUVmax, and SUVmean) is higher in patients with malignant SPNs than in benign ones [21]. In the same study, it has been demonstrated that the addition of shape and texture features increase the performance of both the CT-based and PET-based prediction models. In this context, many radiomic works [22,23,24] help the clinician to correctly formulate the diagnosis. In a study based on dual-time FDG-PET/CT, the authors observed that an improvement in discriminating benign from malignant nodules can be achieved by adding neighborhood gray tone difference matrix texture features to SUVmax and visual interpretation [25]. The future perspective of the current work will be to apply artificial intelligence to FDG-PET/CT studies.

Further consistency in our data is given by the fact that all patients underwent examination using the same PET/CT scanner and acquisition protocol. As a result, both the qualitative and semiquantitative parameters taken into account and obtained in this study were not affected by operator-dependent bias. However, the study has some limitations, due to the retrospective nature of the work and the relatively small cohort of patients. Unfortunately, the smaller sample size makes it more difficult to correlate the morphological characteristics of the nodules with semiquantitative data (it was only observed how the dimension of the lesion at the CT scan directly correlates with SUVmax). On the other hand, data concerning SUVmax, SUVmean, TLG, and MTV seem to be consistent with previous studies and, considering the paucity of studies involving TLG and MTV in the assessment of SPN malignancy at PET/CT, they can offer an additional instrument in the evaluation of SPNs.

## 5. Conclusions

This study showed that qualitative and semiquantitative analysis of SPNs at ^18^F-FDG PET/CT can be very important in assessing the probability of malignancy. It has been observed that in 90.24% of cases, lesions with a high-intensity uptake of ^18^F-FDG at PET/CT are subsequently found to be malignant. Cut-offs for SUVmax (3.625), SUVmean (2.51), MTV (2.55), and TLG (11.8) show good sensitivity and specificity. TLG is directly correlated to SUVmax and SUVmean and can be used as a predictive parameter for malignancy. Nodule size correlated significantly with the intensity of FDG uptake at visual analysis and with SUV max and has proven to be a significant predictor for malignancy. Larger prospective studies could discriminate better between the association between the morphological characteristics of the nodules, radiomic features, and both qualitative and semiquantitative parameters of FDG-PET/CT in order to predict the malignancy of solitary pulmonary nodules.

## Figures and Tables

**Figure 1 cancers-15-01000-f001:**
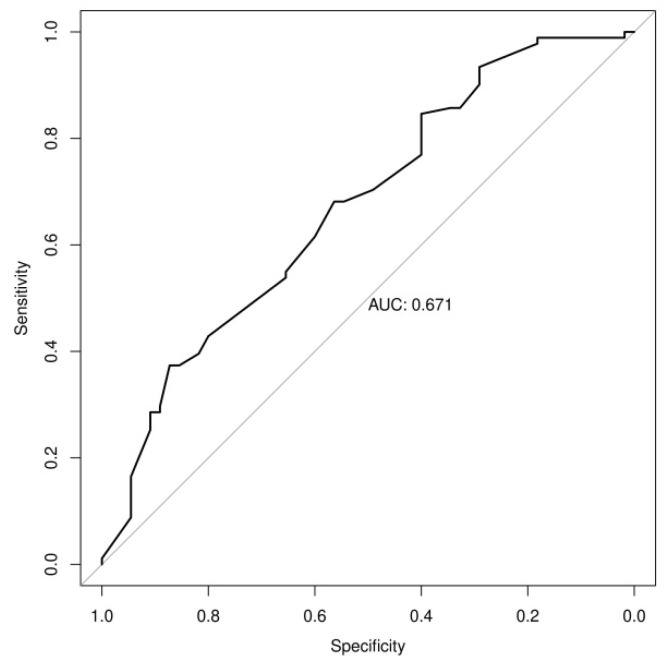
Receiver Operating Characteristic analysis and the related Area Under the Curve for nodule size.

**Figure 2 cancers-15-01000-f002:**
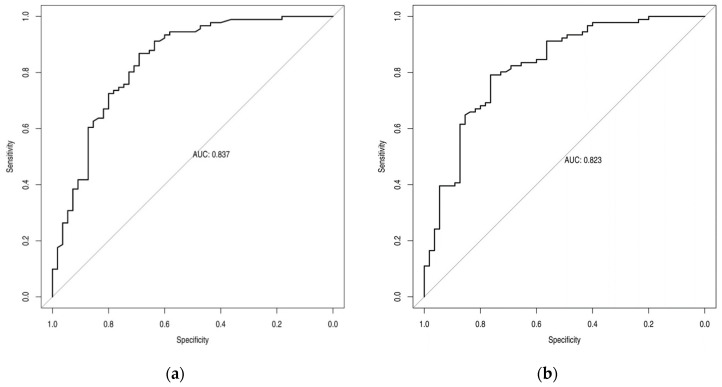
Receiver Operating Characteristic analysis and the related Area Under the Curve for SUVmax (**a**), SUVmean (**b**) MTV (**c**) and TLG (**d**).

**Table 1 cancers-15-01000-t001:** Demographic data and clinical characteristics of the study population.

Characteristics of Patients	Value
Age (years)	68.23 ±8.32
Smoking history	70 (47.95%)
Occupational exposure	18 (12.33%)
Family history	5 (3.42%)
Previous cancer	22 (15.07%)
TBC during childhood	8 (5.48%)
COPD	25 (17.12%)
Pulmonary fibrosis	8 (5.48%)
Emphysema	11 (7.53%)
**Nodule Lung Localization and Size**	**Value**
Superior lobe	74 (50.68%)
Medium lobe	14 (9.59%)
Inferior lobe	58 (39.73%)
Nodule size	18.28 ± 8.32 mm

**Table 2 cancers-15-01000-t002:** Definitive diagnosis of the 146 SPNs.

Histology	Value
Primary lung carcinomas	91 (62.33%)
Adenocarcinoma	63 (69.23%)
Spinocellular carcinoma	11 (12.09%)
Squamous carcinoma	6 (6.59%)
Neuroendocrine tumor	8 (8.79%)
Rare histotypes	3 (3.30%)
Benign lesions	55 (37.67%)

**Table 3 cancers-15-01000-t003:** Visual analysis results for solitary pulmonary nodules (SPNs).

	Overall Population (n = 146)	Malignant SPNs (n = 91)	Non Malignant SPNs (n = 55)
Absent/faint uptake	20	4	16
Moderate uptake	44	13	31
Intense uptake	82	74	8

**Table 4 cancers-15-01000-t004:** Median (MAD) values and optimal cut-offs of maximum standardized uptake value (SUVmax), mean standardized uptake value (SUVmean), metabolic tumor volume (MTV) and total lesion glycolysis (TLG) FDG PET parameters.

	Overall Population (n = 146)	Malignant SPNs (n = 91)	Non Malignant SPNs (n = 55)	Optimal Cut-Offs Values
SUVmax	5.82 (±5.18)	7.89 (±2.24)	2.24 (±1.73)	3.625
SUVmean	3 (±2.12)	3.76 (±2.59)	1.67 (±1.15)	2.51
MTV	2.90 (±2.72)	3.39 (±2.86)	1.20 (±1.20)	2.55
TLG	10.70 (±13.48)	16.36 (±15.87)	1.63 (±2.33)	11.8

**Table 5 cancers-15-01000-t005:** Sensitivity and specificity of maximum standardized uptake value (SUVmax), mean standardized uptake value (SUVmean), metabolic tumor volume (MTV) and total lesion glycolysis (TLG) for optimal cut-offs.

	Sensitivity	Specificity
SUVmax	86.6%	69.1%
SUVmean	79.1%	76.3%
MTV	74.7%	70.9%
TLG	66%	83.6%

## Data Availability

The data presented in this study are available on request from the corresponding author.

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
