# Peer review of "Qualitative and Semiquantitative Parameters of 18F-FDG-PET/CT as Predictors of Malignancy in Patients with Solitary Pulmonary Nodule"

_cancers, 2023, doi:10.3390/cancers15041000_

Round 1
Reviewer 1 Report
In the present study Corica and workers are presenting the use of semi-quantitative parameters 18F-FDG PET/CT for diagnosis of benign and malignant solitary pulmonary nodules. Their conclusions showed that semiquantitative parameters such as SUVmax, SUVmean, TLG and MTV can predict the properties of SPN sensitively and specifically.
1. First of all, some typos are present throughout the text and need to be corrected (i.e. “SUvmean” page 1, line 22, “v” should be capitalized);
2. Note that the “18” in “18F-FDG” should be superscripted (18F-FDG);
3. The author stated in the article that the included patients were all intermediate-risk patients, and risk factors that affect the clinical classification of patients had been mentioned below, however, the authors did not give readers a clear classification criteria;
4. In line 107, “Qualitative analysis of FDG uptake in SPNs was made according to a visual 4-point scoring system, as 1= intense, 2 = moderate, 3 = low, and 4 = absent.”, the authors still did not show a clear standard.
5. All of the data and results were expressed using the mean ± SD, however, did the authors perform normality and homogeneity of variance tests before?
6. In the results section, I don't understand why the authors stated the same thing twice (line 120~125 and line 139~145). Moreover, in this section, whether the result is based on PET or cytopathology, and if the conclusion is based on cytopathology, what does this false negative and false positive mean? If the conclusion was based on imaging performance, the authors should explain what criteria this conclusion was based on in methods section.
7. In the results section, line 162, “TLG values directly correlate with SUVmax, SUVmean, and MTV”, this is recognized, like you said earlier, TLG was calculated as MTV × SUVmean, this does not require further analysis;
8. Although SPNs are defined as lesions less than 3 cm in diameter, it is well known that due to the resolution of PET and the lesions themselves, PET has little advantage in diagnosing too small nodules (i.e. less than 8 mm), and this part of the lesion has a much lower malignancy risk, and how the authors dealt with these patients;
9. In previous studies, many researchers diagnosed SPNs according to both PET semi-quantitative parameters and CT signs and obtained satisfactory results. In this study, authors only included PET semi-quantitative parameters for research, what are the advantages?
Author Response
Dear Professor,
First of all, I want to thank you for the attention shown and the precious suggestions, which we welcome, which certainly improve the quality and comprehensibility of our work.
All suggested corrections have been made in the revised manuscript in red color (the corrections requested by the second reviewer are in green color), as detailed below:
- In reply to the point 1 comment, we have corrected the typos present throughout the text (see page 1, line 27 of the revised manuscript).
- In reply to the point 2 suggestion, we have superscripted “18” in “18F-FDG” (18F-FDG) present throughout the text (see page 1 line 2, page 2 lines 56 and 74, page 3 line 101, page 9 line 297 of the revised manuscript).
- In reply to the point 3 comment, we have clarified the patient enrollment criteria and the model used for determining the pre-test likelihood of malignancy of nodules. The study was carried out in a selected series of patients with pre-test intermediate likelihood of malignancy. We have added this information in the last paragraph of the Introduction section (lines 75 and 76 of the revised manuscript), and in the Materials and Methods section, patient population subsection (lines 83 and 84 of the revised manuscript). Patients with low or high likelihood of malignancy were excluded. The Brock model was used to assess the pre-test likelihood of malignancy. We have added this information in red color in the Material and Methods section, sub-section “Patient population” (lines 84 and 85 of the revised manuscript). We have also added a new reference (reference number 9 in the revised manuscript) referring to this model (McWilliams, A.; Tammemagi, M.C.; Mayo, J.R.; Roberts, H.; Liu, G.; Soghrati, K.; Yasufuku, K.; Martel, S.; Laberge, F.; Gingras, M., et al. Probability of cancer in pulmonary nodules detected on first screening CT. The New England journal of medicine 2013, 369, 910-919, doi:10.1056/NEJMoa1214726.)
- In reply to the point 4 comment, we have better clarified the score system used for PET images visual analysis. We have used the 4-point visual Herder score system; according to this method, nodule tracer uptake is assessed visually in comparison to mediastinal blood pool. We have added these data in red color in the Material and Methods section, sub-section “18F-FDG PET/CT Acquisition and Analysis” (lines 113-116 of the revised manuscript).
We have also added a new reference (reference number 10 in the revised manuscript) referring to Herder model (Callister, M.E.; Baldwin, D.R.; Akram, A.R.; Barnard, S.; Cane, P.; Draffan, J.; Franks, K.; Gleeson, F.; Graham, R.; Malhotra, P., et al. British Thoracic Society guidelines for the investigation and management of pulmonary nodules. Thorax 2015, 70 Suppl 2, ii1-ii54, doi:10.1136/thoraxjnl-2015-207168.).
- Regarding the comments relative to the statistical data made by the reviewer, thanks for raising this point, we agree.
We have now revised the descriptives. Data are expressed as mean and std. dev. only when appropriate, otherwise we report median and MAD (see Material and Methods section, sub-section “Statistical analysis “, lines 121 and 122 of the revised manuscript). In the latter case this is noted explicitly in the text.
- Repetition in results has been eliminated and the criterion on which the results are based has been clarified by changing the text.
The Results section has been restructured also in the light of the suggestions made by the second reviewer who asked to insert new Tables (green color) and reduce the text. We have also grouped in the same Figure (Figure 1) ROC analysis with AUC of SUVmax (Fig.1a), SUVmean (Fig.1b), MTV (Fig.1c) and TLG (Fig.1d) to make the results of the research clearer.
- In the results section, the statement that TLG correlates with SUV has been removed (see Results section, lines 195-199 of the revised manuscript).
- In reply to the point 8 comment referring to nodule size, thanks for raising this point.
I agree with the fact that nodule size can affect the performance of PET and with the observation that bigger nodules have a higher probability of being malignant.
This point has been raised also by the second reviewer who suggested to test the predictive role of nodule size and to investigate whether a cut-off could be found regarding size.
We have thus performed a logistic regression to test the association between several variables, including size, and the risk of malignity. The results are reported in the lines 143-149 of the revised manuscript. Nodule size has proven to be a significant predictive factor of malignancy with a cut-off of 22.75 mm. A sentence regarding nodule size as a predictive factor has also been added in the Abstract section (page 1, lines 39-40 of the revised manuscript) and in the Discussion section (lines 248 and 249).
- In reply to the point 9 comment, Before PET, all patients included in the study had already performed CT scan (we have added this information in the revised manuscript; lines 81 and 82 of the revised manuscript) and CT findings (nodule size, location, spiculation, part-solid nodule type, emphysema) have been considered together with clinical risk factors, in order to assess the pre-test likelihood of malignancy of nodules according to Brock model. Patients with pre-test intermediate risk of malignancy were selected for this study. The enrollment criteria have been better clarified in the revised manuscript (page 2 line 83-86 of the revised manuscript) and the study was then designed in order to evaluate only the importance of PET-investigated metabolic activity (using both qualitative and 4 semiquantitative parameters) in a homogeneous population at pre-test intermediate risk of malignancy.
Reviewer 2 Report
Dear colleagues,
Thank you very much for giving me the opportunity to review your manuscript I read with great interest. In fact, the management of solitary nodules in clinical routine can be challenging.
However, I believe, that several (methodological) points through the manuscript should be addressed before being re-considered for publication.
1. "Qualitative parameters": My understanding of the presented approach is not reproducible and so questionable. I would suggest either to set up cut-off values ( for example low means SUVmax up to 2.0) or use background activities, as stated in the introduction, as presented in the introduction in accordance with relevant literature. Also, you didnt report the interreader agreement before discussion. A lower interreader agreement in the first place/before discussion would make this approach not practicable in clinical practice.
2. "low risk pre-test probability of malignancy": I understand the concern of having a more homogenous cohort. However, I still do not understand how you rated the patients based on all listed parameters? Did you use any multivariable scale? Please specify (even better based on one patient so the reader understands easily how you rated the included patients).
3. "composition of your cohort". What was the indictions of all 146 PET/CT scans performed. Staging? Treatment response assessment? Follow-up? no oncological indication? There isn't any reference to this. However, I believe, it makes a huge difference in the work-up of lung nodules. Also, I found it quite interesting that all of them got a cytopathological examination and none of them regardless of size a follow up with imaging, which is in a clinical setting quite unusual. Also I was suprised by the proportion of primary malignancies, which rises again the question of the indication of the performed scans.
4. I believe there is a significant part in your methodological part missing. Since you used hybrid imaging for your investigations, an more extensive assessment of the morphology of the considered nodules would have been useful too, such as size, margin, solid/subsolid. In fact, all these parameters together also do play a significant role in the management of lung nodules alongside. Also, I would explore the hypothesis the bigger the nodule, the higher the likelihood of malignancy and investigate whether a cut-off could be found with regards to size. Please specify and confront the results with relevant literature.
4. Some improvements in the section results would provide more clarity for the reader. I found it sometimes different to remember everything in the text. I would suggest to add further tables with all results (including histopathology, all morphological criteria, for example for the all cohort first and subsequently in the same table patients with malign findings vs. benign findings.
5. In the discussion part, please report all the new/required insights above all with regards to morphology in light of relevant literature.
6. Please be more specific in the description of your figures.
Looking forward to the revised version.
Best
Author Response
Dear Professor,
First of all, I want to thank you for the attention shown and the precious suggestions, which we welcome, which certainly improve the quality and comprehensibility of our work.
All suggested corrections have been made in the manuscript in green color (the corrections requested by the first reviewer are in red color) as detailed below:
1)In reply to the point 1 comment (qualitative parameters), visual analysis is a method currently used in PET images interpretation. However, to ensure reproducibility and avoid low inter-reader agreement it is important to use a standardized method. In the revised manuscript we have better clarified the score system used for PET images visual analysis. We have used the 4-point visual Herder score system; according to this method, nodule tracer uptake is assessed visually in comparison to mediastinal blood pool. We have added these data in red color in the Material and Methods section, sub-section “18F-FDG PET/CT Acquisition and Analysis” (lines 113-116 of the revised manuscript). We have also added a new reference (reference number 10 in the revised manuscript) referring to Herder model. (Callister, M.E.; Baldwin, D.R.; Akram, A.R.; Barnard, S.; Cane, P.; Draffan, J.; Franks, K.; Gleeson, F.; Graham, R.; Malhotra, P., et al. British Thoracic Society guidelines for the investigation and management of pulmonary nodules. Thorax 2015, 70 Suppl 2, ii1-ii54, doi:10.1136/thoraxjnl-2015-207168.).
Regarding, finally, the inter-reader agreement, it was high. All examination had been performed at the same Nuclear Medicine Unit and examined by two well-trained nuclear medicine physician of the same Unit that resolve disagreements by consensus, as already reported in the original manuscript.
2) In reply to the point 2 comment (pre-test probability of malignancy, homogeneous cohort..), we have clarified the patient enrollment criteria and the model used for determining the pre-test likelihood of malignancy of nodules. The study was carried out in a selected series of patients with pre-test intermediate likelihood of malignancy. We have added this information in the last paragraph of the Introduction section (lines 75 and 76 of the revised manuscript), and in the Materials and Methods section, patient population subsection (lines 83 and 84 of the revised manuscript). Patients with low or high likelihood of malignancy were excluded. The Brock model was used to assess the pre-test likelihood of malignancy. We have added this information in red color in the Material and Methods section, sub-section “Patient population” (lines 84 and 85 of the revised manuscript). We have also added a new reference (reference number 9 in the revised manuscript) referring to this model (McWilliams, A.; Tammemagi, M.C.; Mayo, J.R.; Roberts, H.; Liu, G.; Soghrati, K.; Yasufuku, K.; Martel, S.; Laberge, F.; Gingras, M., et al. Probability of cancer in pulmonary nodules detected on first screening CT. The New England journal of medicine 2013, 369, 910-919, doi:10.1056/NEJMoa1214726.)
3) In reply to the point 3 comment (composition of cohort, FDG PET indications, gold standard, rate of malignant nodules ..), we have clarified the reason of FDG PET scan (metabolic activity evaluation of the nodule) and the criteria used for patient enrollment (pre-test intermediate risk of malignancy calculated according to Brock model) in the revised manuscript (Material and Methods section, Patient population subsection, lines 82-86). Regarding the gold standard used for definitive diagnosis (cytopathological examination) also other studies on SPNs reported in literature are based on this method (one of these studies is cited in the current manuscript; see reference 14, Erdogan et. al). Regarding, finally, the proportion of malignancy, it is that observed following cytopatological examination.
4-5) In reply to the reviewer's point 4 and 5 comments, the study was designed to evaluate only the importance of PET-investigated metabolic activity in patients with SPNs at intermediate likelihood of malignancy. The comparison between morphology and PET was not objective of our study and consequently no specific references regarding morphology are reported in the manuscript.
However, before PET, all patients included in the study had already performed CT scan (we have added this information in the revised manuscript, see lines 81 and 82 of the revised manuscript) and CT findings (nodule size, location, spiculation, part-solid nodule type, emphysema) have been considered together with clinical risk factors, in order to assess the pre-test likelihood of malignancy of nodules according to Brock model.
Regarding to the hypothesis of testing the predictive role of nodule size, also investigating whether a cut-off could be found regarding size, we have performed a logistic regression analysis to test the association between all variables listed in Table 1, including nodule size and nodule location, and the risk of malignity. The results are reported in the lines 143-149 of the revised manuscript: nodule size has proven to be a significant predictive factor of malignancy with a cut-off of 22.75 mm.
A sentence regarding nodule size as a predictive factor of malignancy has also been added in the Abstract section (page 1, lines 39-40 of the revised manuscript) and in the Discussion section (lines 248 and 249).
Furthermore, I agree with the suggestion of improving the Results section. In particular, we have restructured this section, reducing the text and including 4 new Tables. These Tables refer to histopathology (Table 2) and to visual (Table 3) and semiquantitative (Table 4) results in the overall population and in patients with malignant and non-malignant findings, as suggested. In the last new Table (Table 5), sensitivity and specificity values of semiquantitative parameters derived by using the optical cut-offs are reported.
We have also grouped in the same Figure (Figure 1) ROC analysis with AUC of SUVmax (Fig.1a), SUVmean (Fig.1b), MTV (Fig.1c) and TLG (Fig.1d).
We hope that following all the above changes, the Results are clearer.
6) A specific description of Tables and Figure has been added.
Round 2
Reviewer 2 Report
Dear colleagues,
Thank you for the opportunity to review the revised version of your manuscript.
I was pleased to notice the significant improvement of the quality of your manuscript, particularly with regards to results presentation.
I do have final minor recommendations:
One of the main differences between your results and those of the reference manuscript cited is that SUV did not correlate with the nodule size.
So, could you provide the descriptive statistics of lung nodule size, as you did in the new edited table 4. Also without the descriptive statistics it is difficult for the reader to estimate whether the cited cut-off value (22.75 mm) is plausible or not.
According to your discussion this seems to be one the main insights of your investigations. This should also be displayed in the results section with descriptive statistics, eventually also the ROC-curve used for the cut-off value calculation.
No further suggestions since the remaining concerns have been correctly addressed.
Best regards.
Author Response
Dear Professor,
First of all, I want to thank you again for the attention shown and the precious suggestions in the revision of the manuscript, which we welcomed, which certainly improved the quality and comprehensibility of our work.
The last suggested corrections have been made in the second version of the revised manuscript in blue color, as detailed below. Other changes have been made accordingly to the Editor invitation of extending the main text, and are in pink color.
1) In replay to your request of providing “the descriptive statistics of lung nodule size, as did in the new edited table 4” , we have added nodule size of the two subpopulations of nodules, malignant and non malignant, in the Result section (blue color; page 4, lines 146 and 147). Nodule size of the overall population had already been indicated in Table 1.
2) In reply to your request to display in the result section “eventually also the ROC curve” relative to nodule size for the best cut-off value calculation, we have added the curve (Results section, page 4, Figure 1). Following the addition of this figure, the previous Figure 1 became Figure 2 in the revised (see page 7 of the revised manuscript).
3) Finally, we would like to thank you for the careful revision of the manuscript. Your comment on SUVmax and the absence of correlation with the nodule size, fortunately allowed us to discover the error reported in the text.
At statistical analysis, nodule size correlated significantly with SUVmax (r= 0.419, p<0.001) and also with the intensity of FDG uptake at visual analysis (r= 0.338, p<0.001).
We have added these data extensively in the Results section (page 4, lines 148-150). A little phrase regarding this has also been added in the abstract section (lines 39-40) and in the Conclusions section (page 9, lines 331-332). In the Discussion section, we have inserted the correct information (page 8, lines 263-264 and page 9 lines 291-292 of the revised manuscript) and deleted the errors (see lines 265-266 and lines 292-294).
Thank you again for your very constructive comments.
Best regards,
Viviana Frantellizzi and co-workers